# Efficient method for isolation of high-quality RNA from *Psidium guajava* L. tissues

**Paola de Avelar Carpinetti**[☉]**, Vinicius Sartori Fioresi**[☉]**, Thais Ignez da Cruz, Francine Alves Nogueira de Almeida**[iD]**, Drielli Canal, Adésio Ferreira, Marcia Flores da Silva Ferreira**[iD] *

Laboratory of Genetics and Plant Improvement, Department of Agronomy, Centre for Agricultural Sciences and Engineering, Federal University of Espírito Santo, Alegre, ES, Brazil

☉ These authors contributed equally to this work.
* marcia.ferreira@ufes.br

**Data Availability Statement:** All relevant data are within the manuscript and its Supporting Information files.

**Funding:** The author(s) received no specific funding for this work.

## Abstract

Acquiring high-quality RNA in sufficient amounts is crucial in plant molecular biology and genetic studies. Several methods for RNA extraction from plants are available in the literature, mainly due to the great biochemical diversity present in each species and tissue, which can complicate or prevent the extraction. *Psidium guajava* (Myrtaceae family) is a perennial fruit tree of medicinal and economic value; nevertheless, only a few molecular studies are available for the species. One reason is the difficulty in obtaining RNA due to the content of the samples, which are rich in polyphenols, polysaccharides, and secondary metabolites. Furthermore, there are few studies available for the isolation of RNA from guava or *Psidium* samples, which hampers advances in the study of the genus. Here, quality and yields of RNA isolates were compared using six extraction protocols: two protocols based on the application of cetyltrimethylammonium bromide (CTAB) lysis buffer, one protocol which uses the TRIzol reagent, one which applies guanidine thiocyanate lysis buffer followed by organic phase extraction, and two commercial kits (PureLink RNA Mini Kit and RNeasy Plant Mini Kit). The CTAB-based method provided the highest RNA yields and quality for five different tissues (flower bud, immature leaf, young leaf, mature leaf, and root), genotypes, and stress conditions. For the most efficient protocol, the average yield of RNA from guava leaves was 203.06 µg/g of tissue, and the $A_{260}/A_{280}$ and $A_{260}/A_{230}$ ratios were 2.1 and 2.2, respectively. RT-qPCR analysis demonstrated that the purity of the samples was sufficient for molecular biology experiments. CTAB-based methods for RNA isolation were found to be the most efficient, providing the highest RNA yields and quality for tissues from *P. guajava*. Additionally, they were compatible for downstream RNA-based applications, besides being simple and cost-effective.

## Introduction

Advances in cell and molecular biology–such as the discovery of genes, underlying mechanisms of gene regulation, signal transduction, and the factors involved in phenotypic characteristics–require a whole range of techniques, such as northern hybridization, reverse transcription, polymerase chain reaction (PCR), construction of cDNA libraries, and

**Competing interests:** The authors have declared that no competing interests exist.

sequencing. Obtaining high RNA quality and yields is fundamental for the execution and success of these approaches. The isolation of intact RNA is difficult due to the chemical nature of the RNA molecule, which is more susceptible to hydrolysis, and its sensitivity to enzymatic degradation by ribonucleases (RNases), which are very active, widespread, stable, and require no cofactors [1, 2]. Furthermore, the isolation and purification of high-quality RNA from plant tissues in sufficient amounts has been reported to be difficult in samples rich in polyphenols, polysaccharides, and other secondary metabolites that interfere with the quantification and subsequent applications [3–7].

Guava (*Psidium guajava* L., family Myrtaceae) is an important commercial fruit crop cultivated in tropical and subtropical regions of the world. The largest producers are India, China, Thailand, Pakistan, Mexico, Indonesia, and Brazil, respectively [8]. Different parts of the plant are rich in nutrients and functional elements, such as antioxidants, vitamin C, potassium, and fiber [9, 10]. It is also widely used for medicinal purposes due to its healing, anti-allergic, anti-diabetic, anti-diarrheal, anti-neoplastic, anti-inflammatory, and anti-microbial properties [11–17]. Although relevant knowledge is available about the species' biochemical composition, benefits, and applications, when compared to other crops, genetic and genomic studies in guava are incipient. Developments in the genetic analysis and functional genomics of guava could complement the conventional breeding process, leading to improvement of crop productivity and addressing the challenges of enhancing fruit quality and tolerance to abiotic and biotic stresses [18–20]. However, the success of many downstream RNA-based applications relies on obtaining high-quality RNA.

The use of cetyltrimethylammonium bromide (CTAB) to isolate nucleic acids was originally described in 1953 for bacterial samples [21]. Since then, this protocol has been extensively used and modified for the extraction of DNA and RNA from several species [22–26]. In recent years, CTAB-based methods have been used for nucleic acid isolation from many plant species, especially those containing a high level of phenolic compounds and polysaccharides [3, 27–29]. Most of the popular RNA isolation protocols are based on guanidine salts, such as the reagent TRIzol (Invitrogen, Carlsbad-CA–USA) and commercial kits like RNeasy (QIAGEN, Hilden–Germany) and PureLink RNA (Invitrogen, Carlsbad-CA–USA). Although these methods have been used successfully for the isolation of RNA from tissues for a variety of plants [30–34], for certain species, protocols based on the guanidine method have proven to be unable to isolate high quality RNA with satisfactory yields; moreover, they increase the chances of co-purifying contaminants, which interfere with downstream applications [7, 35, 36].

The present study was motivated by the lack of consensus concerning which methods are more efficient for the extraction of sufficient quantities of high-quality RNA from *P. guajava*; mainly for use in studies of gene expression and omics analyses. Here, two protocols that use a buffer based on a cationic surfactant, CTAB, and four methods using chaotropic lysis buffers containing guanidine thiocyanate were examined. In the guanidine group, two commercial kits that use silica membrane to isolate RNA were evaluated, PureLink RNA Mini Kit and RNeasy Plant Mini Kit. The third method is based on the association of lysis buffer with phenol-chloroform, the TRIzol reagent. The last protocol in this group is the guanidine thiocyanate method and it is the only one already described for the isolation of *P. guajava* RNA [37]. The goal of this study was to identify and/or perfect an efficient RNA isolation method for various *P. guajava* tissues to be used in plant molecular biology studies.

## Results and discussion

RNA isolation plays an important role in various experiments in plant molecular biology, such as gene expression and transcriptomics. However, obtaining RNA with enough quality and

**Fig 1. Qualitative analysis of RNA by agarose gel electrophoresis.** Denaturing agarose gel (1%) electrophoresis of total extracted RNA (2μL) stained with GelRed (Biotium, Fremont-CA–USA). For the positive control (C+), total RNA was extracted from soybean (*Glycine max*) leaves, and extractions using *P. guajava* leaves were performed in duplicate (R1 and R2).

quantity may be a great challenge, mainly for plant species since their biochemical composition can hinder and even prevent the extraction. Moreover, due to the rich variety of cellular chemical contents of plants, there are no standard methods for the isolation of RNA applicable to all plant species. Members of the Myrtaceae family are characterized by an abundance of essential oils rich in terpenes, tannins, phenolic compounds, polysaccharides, and other secondary metabolites that hinder the extraction of nucleic acids from these species [10, 38, 39]. This highlights the need to seek an efficient, effortless, and cost-effective extraction method to contribute to the advances in molecular studies for the species of this family.

In this study, six RNA isolation methods were compared, and the success of each protocol was judged by the quantity, purity, and integrity of the recovered RNA. Fig 1 shows the outcomes for RNA integrity, assessed by agarose gel electrophoresis. For $CTAB_1$ and $CTAB_2$ as well as the modified Salzman *et al.* [40] protocols, distinct 25S and 18S rRNA bands were visible, with high brightness and no obvious smearing due to degradation, suggesting that these methods ensured RNA integrity. On the contrary, for the samples prepared with TRIzol, the RNeasy Plant Mini Kit and the PureLink RNA Mini Kit protocols, no RNA bands appeared, nor was degradation smear observed. The $A_{260}/A_{280}$ and $A_{260}/A_{230}$ ratios are often used to indicate RNA sample purity, with ratio values of 2.0–2.2 generally indicating high RNA purity.

The values for $A_{260}/A_{280}$ ratio (Table 1) for $CTAB_1$ (2.10 ± 0.04) and $CTAB_2$ protocols (2.13 ± 0.04) indicated that these methods were efficient in preventing protein contamination. On the other hand, for the TRIzol method, RNeasy Plant Mini Kit and the PureLink RNA Mini Kit protocols, the low values indicate that there were problems in the extraction and a substantial amount of proteins precipitated with the nucleic acids. Importantly, regarding the $A_{260}/A_{230}$ ratio, only CTAB-based protocols showed values that correspond to high purities (2.24 ± 0.33 and 2.24 ± 0.07). The other samples showed high absorbance at 230 nm, a wavelength at which carbohydrates, phenols, and aromatic compounds generally absorb.

**Table 1. Analysis of RNA yield and purity for different methods.**

| Method | n | Yield (ng/μL) | | | $A_{260}/A_{280}$ ratio | | | $A_{260}/A_{230}$ ratio | | |
|---|---|---|---|---|---|---|---|---|---|---|
| $CTAB_1$ | 6 | 386.28 [a] | ± | 69.61 | 2.10 [a] | ± | 0.04 | 2.24 [a] | ± | 0.33 |
| $CTAB_2$ | 6 | 676.87 | ± | 191.96 | 2.13 [a] | ± | 0.04 | 2.23 [a] | ± | 0.07 |
| PureLink RNA Kit | 5 | 120.40 [a] | ± | 32.13 | 1.00 | ± | 0.20 | 0.23 | ± | 0.06 |
| RNeasy Plant Kit | 6 | 35.52 | ± | 7.38 | 1.22 | ± | 0.14 | 2.11 | ± | 0.05 |
| TRIzol reagent | 6 | 444.45 [a] | ± | 134.98 | 1.09 | ± | 0.13 | 0.49 | ± | 0.10 |
| Modified Salzman *et al.* | 6 | 436.70 [a] | ± | 257.73 | 1.88 [a] | ± | 0.14 | 1.39 | ± | 0.26 |

The values correspond to the mean of the values and MAD (mean absolute deviation) obtained between the replicates (n) of each method in two independent experiments. Values within rows followed by the same letter are not significantly different, using One-way ANOVA followed by Dunnett's test, $p < 0.05$ when compared with $CTAB_2$.

**Table 2. Comparison of RNA extraction methods.**

| Method | Processing time | Maximum spin speed required | Efficiency | Advantages | Disadvantages |
|---|---|---|---|---|---|
| CTAB$_1$ | ~18 hours (including incubation) | 25,000 ×$g$ | Good yield and high purity | High efficiency in RNA isolation and less presence of PCR inhibitors. Inexpensive. | User-made buffers need more caution to avoid contamination. Longer procedure (overnight incubation and multiple transfers between tubes). High-speed spin required. |
| CTAB$_2$ | ~6 hours (including incubation) | 16,000 ×$g$ | High yield and high purity | High efficiency in RNA isolation. User-friendly procedures. Inexpensive. | User-made buffers need more caution to avoid contamination. |
| PureLink RNA Kit | less than 1 hour | 12,000 ×$g$ | Low or no yield, low integrity, and low purity | Rapid protocol with few steps. Kit method with all buffers provided (avoids contamination). | More expensive than non-kit methods. Greater difficulty in adapting the method upon the inefficiency of the RNA extraction. |
| RNeasy Plant Kit | less than 1 hour | 10,000 ×$g$ | Low or no yield, low integrity, and low purity | Rapid protocol with few steps. Kit method with all buffers provided (avoids contamination). | More expensive than non-kit methods. Greater difficulty in adapting the method upon the inefficiency of the RNA extraction. |
| TRIzol reagent | 1–1.5 hour | 12,000 ×$g$ | Low or no yield, low integrity, and low purity | Simple protocol with few steps and reagents. Very efficient lysis (prevents degradation of RNA). | No visible bands in the agarose gel. Residual contaminants (possibly phenol and salts). |
| Modified Salzman *et al.* | 4–5 hours (including incubation) | 16,000 ×$g$ | Low yield, and good integrity | User-friendly procedures and less expensive. | User-made buffers need more caution to avoid contamination. Low efficiency and possible residual contaminants. |

The results of the qualitative analyses (Fig 1) are consistent with the quantifications obtained by spectrophotometry shown in Table 1. The RNA concentrations of the samples were obtained by evaluating the absorbance at 260 nm ($A_{260}$). In this analysis, the RNA samples from CTAB$_2$ had the highest yields, 203.06±57.6 µg of RNA for each gram of leaf tissue. CTAB$_1$ and the modified Salzman *et al.* methods also provided good yields (115.88 ± 20.88 and 131.01 ± 77.32, respectively). However, the yield of the guanidine-based method is possibly inaccurate, since the intensity of the rRNA bands is weaker than in the CTAB$_1$ method. This may be a reflection of contaminants present in modified Salzman *et al.* protocol samples, such as the guanidine salt, evidenced by the low 260/230 ratio (1.39 ±0.26), or genomic DNA contamination. The contaminants also absorb in the 260 nm range, thus possibly distorting the quantification. The same occurred in samples extracted with TRIzol, which showed great absorption at $A_{260}$, but presented very low $A_{260}/_{280}$ and $A_{260}/_{230}$ ratios (1.09 ± 0.13 and 0.49 ± 0.10) and no visible bands in the agarose gel. Additionally, both evaluated commercial kits failed to isolate RNA from *P. guajava* leaves, as demonstrated by the poorer yields obtained and the analysis of $A_{260}/_{280}$ and $A_{260}/_{230}$ ratios indicating a significant contamination of the samples.

CTAB-based methods were more efficient for the isolation of RNA from guava leaves, considering the yield, integrity, and purity. However, CTAB$_2$ was the only method with which it was possible to obtain statistically significant high yields. Table 2 summarizes the main parameters evaluated in this study to compare the six methods for isolating RNA from *P. guajava* leaves, as well as crucial issues for choosing an RNA isolation method, such as processing time, cost, and method complexity, among others. As seen in Tables 1 and 2, the CTAB$_2$ method was preferred and used for more detailed evaluations.

These findings are compatible with other studies which demonstrated that methods based on guanidine salts, TRIzol, and commercial kits were not effective for extracting RNA from many species rich in secondary metabolites [4, 7, 35]. The CTAB-based methodology has also been used successfully by Jaakola *et al.* (2001) for the extraction of high-quality RNA from *Vaccinium myrtillus* [41], and by Zeng & Yang (2002) in *Cinnamomum tenuipilum* [29], both

perennial tree species. More recently, researchers studying species from the Myrtaceae family have isolated RNA by the CTAB method and performed analyses of transcriptome and gene expression, such as Guzman *et al.* (2014) in *Eugenia uniflora* [42], and Vining *et al.* (2015) and Favreau *et al.* (2019) in *Eucalyptus grandis* [43, 44]. In the study involving *P. guajava*, Furlan *et al.* (2012) isolated RNA using the guanidine thiocyanate buffer evaluated in this study [37]. However, in the comparative analysis, this method did not show the best results.

Guava samples have a high content of polyphenols, polysaccharides, tannins, among other metabolites that may interfere with the RNA isolation [45, 46]. In solution, phenolic compounds are easily oxidized to form quinones, which can bind to the RNA and render it insoluble, hindering the RNA isolation and/or downstream applications [46]. During extraction, polysaccharides produce saccharide fragments that may be co-purified with the RNA, owing to their chemical characteristics being very similar to nucleic acids [45, 47]. The greater efficiency of the extraction by the CTAB method may be related to the composition of the respective lysis buffer. CTAB is a cationic surfactant, acting as a strong detergent to help break membranes and separate the nucleic acids from polysaccharides and cellular debris [48]. The oxidation of polyphenols is prevented by the use of soluble polyvinylpyrrolidone polymers (PVP) that immediately bind to those compounds, and this precludes the formation of quinones and their subsequent binding to RNA [41]. Also, heating of the extraction buffer with a relatively high concentration of β-mercaptoethanol promotes more efficient tissue disruption, removal of tannins and polysaccharides, and prevention of RNase activity [49, 50]. Carbohydrate contamination is reduced by the use of LiCl in the precipitation step, since the saccharides remain in solution while the RNA precipitation occurs [49].

It is known that the biochemical composition varies greatly between tissues, and alters even more in response to some conditions of stress, owing to the physiological metabolism and remodeling of pathways for the stress response. Moreover, isolating RNA of adequate quality from tissues exposed to stressful conditions such as salinity and heavy metal toxicity is difficult due to increased accumulation of reactive oxygen species (ROS), secondary metabolites, and other compounds that easily degrade nucleic acids or promote chemical changes in nucleic acids that hinder isolation [51, 52]. Aiming to expand the analysis and verify the usefulness of the proposed protocol, the ability of the CTAB$_2$ protocol to extract RNA from different *P. guajava* tissues (Fig 2 and Table 3) was also evaluated from tissues under salt and cadmium stress conditions (Table 3), and from another species of the same genus, *Psidium guineense* (S1 Fig).

In summary, the proposed protocol provided high RNA yields, quality, and purity in the assessed tissues and conditions. In this study, satisfactory yields and quality were observed even in developing tissues, such as flower buds, which usually accumulate high amounts of secondary metabolites and have been reported for poorer quality [53]. The tissues with the highest yield were flower buds, immature leaves, and young leaves, respectively. This result is possibly related to their metabolically more active tissues. Mature leaves, in addition to having less active cell metabolism and senescent cells, may already contain cells in the process of cell death, which may explain the lower yields. Root tissues, on the other hand, are commonly metabolically active, but their cellular water content is greater than that of leaf tissues, resulting in a lower amount of total RNA per gram of tissue. Among the different genotypes of *P. guajava*, no significant differences were observed regarding the RNA yields obtained, demonstrating the potential efficiency of this protocol for other genotypes of this species.

These findings were demonstrated mainly by the analysis using TapeStation capillary electrophoresis (Fig 2B), in which these samples displayed an RIN (RNA Integrity Number) [54] range between 8.8 and 5.0. Unlike mammalian tissue, in which high quality RNA has an RIN > 9, these values are not possible for plant RNA. Plants have an abundance of other rRNA subunits (5S, 8S, 16S, 18S, 23S, and 25S) from cytosolic, chloroplastic, and

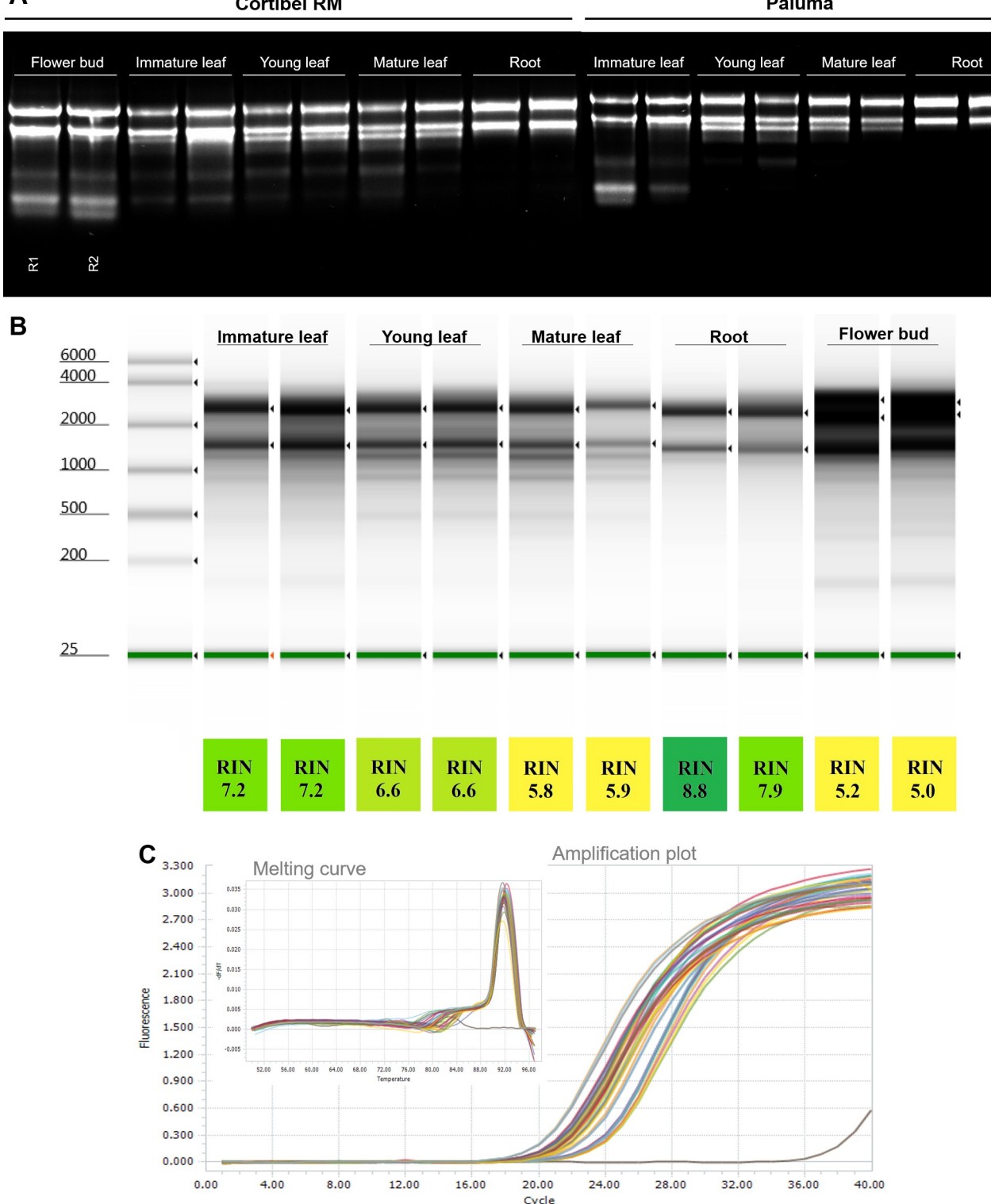

**Fig 2. Qualitative analysis of RNA from different tissues. (A)** Denaturing agarose gel (1%) electrophoresis of total extracted RNA (2μL) stained with GelRed (Biotium, Fremont-CA–USA). Extractions using the $CTAB_2$ protocol were done in duplicate (R1 and R2) for different tissues from two distinct genotypes of *P. guajava*, Cortibel RM and Paluma. **(B)** Quality analysis of RNA samples from Cortibel RM cultivar by TapeStation (Agilent Technologies, Santa Clara-CA–USA) using RNA ScreenTape assay, after DNase I treatment, and RIN (RNA Integrity Number) were calculated. The analyses were performed with two samples of each tissue: immature leaf, young leaf, mature leaf, root, and flower bud, respectively. **(C)** Quality check

of cDNA from CTAB$_2$ samples (Cortibel RM tissues) by amplification of histone H2A gene measured using SYBR Green RT-qPCR, and melting curve showing specific amplification products.

mitochondrial compartments, generating some degree complexity in the RIN reading, because these additional peaks are mistaken for rRNA degradation. In addition, green tissue may contain additional rRNAs in contrast to non-green tissues and have lower and sometimes indeterminate RIN values. Therefore, the values obtained are compatible with high quality RNA in plants, and as expected, the RIN values are higher in roots, due to the greater abundance of cytosolic rRNA (18S and 25S) and the absence of chloroplast ribosomes (5S, 16S, and 23S) [55, 56].

Molecular studies, such as RNA-Seq, usually use RNA samples with at least RIN > 7 to ensure sequencing quality. However, as previously discussed, even for samples from chlorophyll-containing tissues with high quality and integrity, the RIN may not be correctly calculated, and in these cases, other parameters have to insured, as observed for RNA experiments using formalin-fixed paraffin-embedded (FFPE) tissues. For FFPE tissues, which is a low-quality source (RIN usually ranging from 2 to 5), the chance of success is based upon factors such

**Table 3. Analysis of RNA yield and purity using CTAB$_2$ method.**

| RNA yield and purity for different tissues | | | | | | | | | | | |
|---|---|---|---|---|---|---|---|---|---|---|---|
| Tissue | | n | Yield (ng/µL) | | | A$_{260}$/A$_{280}$ ratio | | | A$_{260}$/A$_{230}$ ratio | | |
| Cortibel RM | Flower bud | 4 | 1972.28 [a] | ± | 642.49 | 2.00 [c] | ± | 0.17 | 2.26 [b] | ± | 0.07 |
| | Immature leaf | 12 | 1072.98 [b] | ± | 184.14 | 2.15 [ab] | ± | 0.04 | 2.21 [b] | ± | 0.08 |
| | Young leaf | 12 | 753.76 [c] | ± | 64.63 | 2.14 [b] | ± | 0.01 | 2.33 [b] | ± | 0.05 |
| | Mature leaf | 12 | 566.32 [d] | ± | 93.23 | 2.11 [b] | ± | 0.04 | 2.40 [b] | ± | 0.07 |
| | Root | 12 | 613.91 [cd] | ± | 118.68 | 2.19 [a] | ± | 0.03 | 2.55 [a] | ± | 0.35 |
| Paluma | Immature leaf | 12 | 1038.41 [a] | ± | 281.16 | 2.14 [b] | ± | 0.03 | 2.24 [b] | ± | 0.10 |
| | Young leaf | 12 | 742.11 [b] | ± | 113.15 | 2.17 [ab] | ± | 0.04 | 2.31 [b] | ± | 0.06 |
| | Mature leaf | 12 | 357.80 [c] | ± | 67.82 | 2.22 [a] | ± | 0.03 | 2.32 [b] | ± | 0.07 |
| | Root | 12 | 279.39 [c] | ± | 31.53 | 2.21 [a] | ± | 0.02 | 2.53 [a] | ± | 0.10 |
| RNA yield and purity for stressed leaves from different *P. guajava* genotypes | | | | | | | | | | | |
| Genotype | | n | Yield (ng/µL) | | | A$_{260}$/A$_{280}$ ratio | | | A$_{260}$/A$_{230}$ ratio | | |
| Control | Paluma | 6 | 629.95 [a] | ± | 93.96 | 2.25 [a] | ± | 0.05 | 2.40 [a] | ± | 0.17 |
| | Cortibel LG | 6 | 749.25 [a] | ± | 221.04 | 2.22 [ab] | ± | 0.02 | 2.37 [a] | ± | 0.04 |
| | Cortibel RG | 6 | 622.35 [a] | ± | 31.32 | 2.20 [abc] | ± | 0.01 | 2.36 [a] | ± | 0.01 |
| | Cortibel RM | 4 | 666.59 [a] | ± | 75.36 | 2.21 [abc] | ± | 0.02 | 2.34 [a] | ± | 0.06 |
| | Cortibel XIII | 3 | 805.02 [a] | ± | 409.20 | 2.17 [bc] | ± | 0.02 | 2.24 [a] | ± | 0.07 |
| Salt Stress | Paluma | 6 | 615.00 [a ns] | ± | 48.45 | 2.15 [c] | ± | 0.01 | 2.33 [a] | ± | 0.04 |
| | Cortibel LG | 6 | 633.40 [a ns] | ± | 97.81 | 2.18 [bc] | ± | 0.02 | 2.38 [a] | ± | 0.05 |
| | Cortibel RG | 6 | 601.75 [a ns] | ± | 116.78 | 2.16 [bc] | ± | 0.01 | 2.41 [a] | ± | 0.02 |
| Cd | Cortibel RM | 3 | 204.59 [a **] | ± | 122.02 | 2.09 [a] | ± | 0.08 | 1.88 [a] | ± | 0.34 |
| | Cortibel XIII | 4 | 219.99 [a *] | ± | 65.34 | 2.08 [a] | ± | 0.08 | 2.06 [a] | ± | 0.19 |

The values correspond to the mean of the values and MAD (mean absolute deviation) obtained between the replicates (n) of each condition. Values within rows followed by the same letter are not significantly different, using One-way ANOVA followed by Tukey's test (p<0.05). For stressed samples, the comparison was also made between the control vs. treatment values. ns = not significant

* (p< 0.05)

** p< 0.01).

Cd = Cadmium stress.

as storage time, conditions, fixation time, and specimen size [57]. To enrich this discussion, it should be noted that eighteen samples of different guava tissues, isolated using the $CTAB_2$ protocol, were used for sequencing of total RNA (with depletion of rRNA) on the Illumina platform, out of which, twelve samples were RIN < 7. However, all of them generated high-quality reads, according to the FastQC quality control tool (data not shown).

The $CTAB_2$ protocol showed satisfactory results also in relation to the isolation of RNA for plant tissues under stress (Table 3). The RNA yields obtained demonstrate the efficiency of the protocol, which obtained approximately 185 µg of RNA per gram of tissue from plants under saline stress. For leaf samples from plants under stress, a statistically significant reduction was observed in cadmium stress, when compared with untreated samples (control). However, this result was already expected, since these treatments can induce the degradation of nucleic acids as a result of the triggered cell death as well as the shift in the chemical content [58, 59]. The 260/230 and 260/280 ratios were greater than 2 in most samples, demonstrating high-quality RNA samples.

The presence of contaminants in samples not only hampers the isolation and quantification of the RNA but may also interfere with the activity of some enzymes, such as reverse transcriptase in the synthesis of cDNA and DNA polymerase in PCR. Substances used in extraction techniques, such as ethanol, phenol, chloroform, and salts, are described to inhibit these enzymes and consequently impair these techniques. However, inherent substances in the sample, such as polyphenols and polysaccharides can also co-precipitate with the RNA and affect downstream applications [60–65]. In that regard, to ascertain the quality of the samples for applications in molecular biology, samples of four different tissues (immature leaf, young leaf, mature leaf, and root) were also evaluated via RT-qPCR. All assessed samples showed a characteristic amplification curve and unique peaks in the melting analysis, demonstrating their adequate quality for molecular biology studies (Fig 2C).

In addition, to further investigate the differences between the CTAB protocols, the presence of inhibitors in the samples was inferred by quantitative analysis via RT-qPCR. Linear regression curves based on two-fold serial dilutions containing prepared cDNA synthesized from the same volume of RNA (5 µL) were used for all samples, regardless of their concentration. This was done in order to compare the interference of the contaminants in the samples, in which the sample volume directly affects the inhibitor concentration; different volumes of samples would add another variable into the interpretation of the data. However, despite this condition, on average, 1 µL of cDNA contained 100 ng (± 25 ng). From the cycle threshold (Ct) values obtained by amplification of the Histone H2A gene, a linear equation was generated for each sample. Under ideal conditions, each amplification cycle duplicates the cDNA content, and the efficiency of the reaction is said to be one hundred percent. However, especially for samples obtained from plants, without an additional purification step, the presence of contaminants is very frequent [66].

Fig 3 shows the graphs for all samples. The effect of PCR inhibition is reflected on the distance between the dots and lines (linear equation) and the obtained values (solid lines) as well as the ideal values (dotted lines). From these results, it can be inferred that all samples present some level of DNA polymerase inhibition, as expected for plant samples [67], and the interference is slightly more accentuated in $CTAB_2$ than in $CTAB_1$ samples.

These effects probably reflect competitive inhibition, since sample dilutions were followed by a reduction of interference. It is also worth mentioning that, although the $CTAB_2$ protocol apparently shows a greater amount and/or greater inhibitory effect, it most likely did not hamper the RT-qPCR analysis, which can be explained by a few factors. First, as indicated by the squared correlation coefficient ($R^2$), contamination follows a linear regression. Second, the amount usually applied in the PCR reaction is between 30–10 ng per reaction and, for this

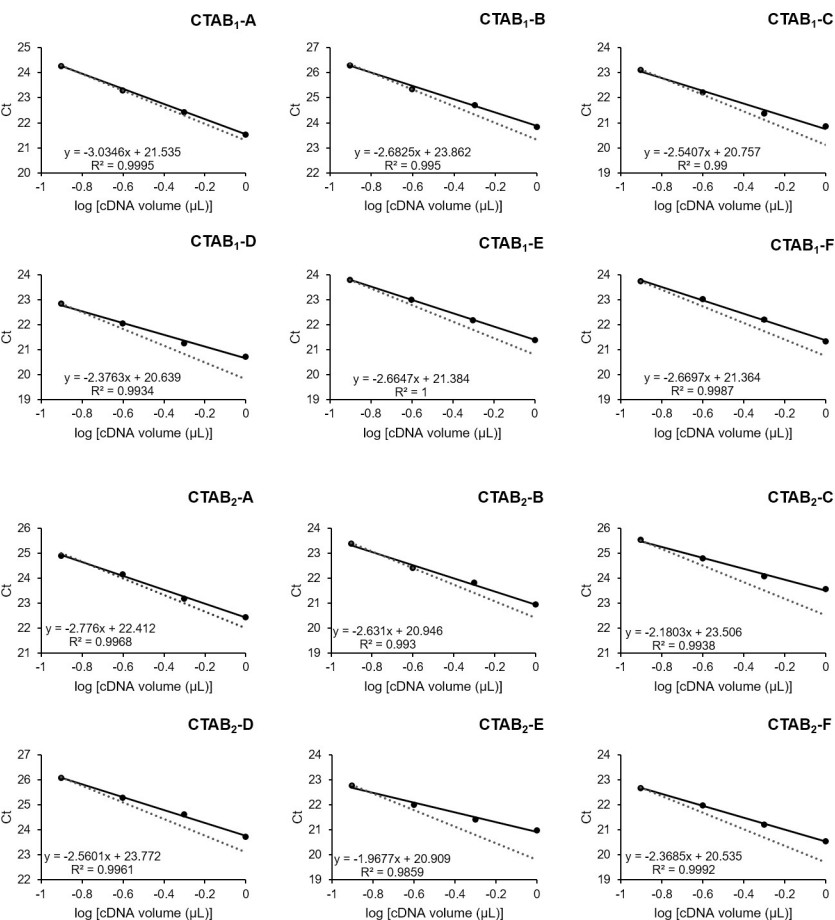

**Fig 3. Quantitative analysis by RT-qPCR to compare CTAB methods.** Linear regression curves of the real-time reverse transcription polymerase chain reaction assay based on serial dilutions of cDNA (1.0, 0.5, 0.25, and 0.125 μL) synthesized from the same volume of RNA (5 μL) were used for all samples. The log of cDNA volume (μL) is indicated on the x-axis, whereas the corresponding Ct values are shown on the y-axis. Each dot represents the mean result of three replicates. The correlation coefficient ($R^2$) and linear equation of the regression analysis are shown. Values obtained in this assay for each sample are represented in solid lines and ideal values in dotted lines, estimated based on the Ct values of the highest dilution, considering an efficiency of one hundred percent in each cycle.

range, the values obtained are close to the ideal value. Lastly, although the samples for this test were selected within the same concentration range (S1 Table), those obtained using $CTAB_2$ were consistently more concentrated, as already discussed and presented in Table 3. Thus, in each PCR reaction, in order to apply the same amount of template, the sample must be further diluted, which will additionally minimize the effect of contaminants.

## Study strength and limitations

Guava trees are part of a family (Myrtaceae) characterized by the large number of species and wide geographical distribution. Although several methods of RNA extraction from plants are available in the literature, very few are dedicated to studies of tree and fruit species, especially involving a large number of samples, variety of tissues, and genotypes, in addition to comparisons under stress conditions. Thus, the establishment of this RNA isolation method capable of extracting high-quality RNA from various tissues may assist research groups that intend to carry out molecular studies with this and other species of the genus, providing an easy-to-

perform method that obtains RNA with a high yield and quality, avoiding unnecessary expenditure of time and financial resources. In this study, the proposed high-quality RNA isolation method does not require high performance centrifuges, is low cost, and saves processing time. Another item that confirms the strength of this study is the RT-qPCR assay, which attests to the quality and potential application of the samples in downstream experiments.

In future studies, it would be interesting to include more commercial kits to the comparison, to expand the analyses of capillary electrophoresis for all samples, as well as to use a method of direct quantification of RNA (such as assays with specific fluorescent probes for RNA). Moreover, further detailed investigations may be conducted applying these samples for the construction of cDNA libraries for sequencing, specifically those from tissues with chlorophyll, due to the challenge of obtaining accurate RIN.

## Conclusion

In the present study, six RNA isolation methods were compared, and CTAB-based protocols were found to be the most efficient, providing the highest RNA yields and quality for different tissues, genotypes, and under different stress conditions. Additionally, it was demonstrated that samples are compatible for downstream RNA-based applications, besides showing the advantages of being faster and more cost-effective. This knowledge contributes to the increase and quality of molecular genetic studies for *Psidium guajava* and related species.

## Materials and methods

### Plant material

Clonal seedlings of *Psidium guajava* (Paluma and Cortibel RM genotypes) were obtained from the Frucafé Nursery, accredited by the Brazilian Ministry of Agriculture. The specimens used in this study were obtained and maintained in healthy conditions in a greenhouse.

For the comparison of protocols, six leaf discs with a diameter of 0.8 cm (70–100 mg) were collected from young leaves of the Paluma genotype when the plants were approximately 100 days old. Each sample represented a pool of tissues obtained from three different plants. All collected biological material was immediately frozen in liquid nitrogen and subsequently stored in an ultra-low temperature (ULT) freezer at -80˚C.

To evaluate the efficiency of RNA isolation in different tissues and genotypes, flower buds, immature leaf, young leaf, mature leaf, and roots from the Cortibel RM genotype and immature leaf, young leaf, mature leaves, and roots from the Paluma genotype were used (S2 Fig). Approximately 100 mg of leaf tissue were used for each sample, except the flower bud and root, for which 80 and 200 mg of tissue were used, respectively.

### RNA isolation

All reagents were prepared in RNase-free ultrapure water, and glasses, mortars and pestles were treated with 1 M NaOH (Neon,Suzano-SP–Brazil), autoclaved, and then baked overnight at 60˚C. At least five extractions were performed for each protocol, as well as two extractions with soybean (*Glycine max*) as a positive control. Soybean tissue was used as a positive control due to the ease of obtaining nucleic acids. In addition, it is a well-studied plant and for most of the methods evaluated in this study, RNA was extracted with high efficiency, including the commercial kits. All samples were extracted starting from the same material and in the same quantity. The tissue samples were ground in liquid nitrogen using a porcelain mortar (60 mL) until they became a fine powder. Care was taken to ensure that the material did not thaw in

this process. The extraction buffer corresponding to each method was added to samples of the obtained tissue powder.

**Cetyltrimethylammonium bromide (CTAB)-based methods.** Extraction buffer: 2% CTAB (Sigma-Aldrich, St. Louis-MO–USA), 2% PVP (mol wt 30,000) (Sigma-Aldrich, St. Louis-MO–USA), 100 mM Tris-HCl (pH 8.0)(Sigma-Aldrich, St. Louis-MO–USA), 25 mM EDTA (Alphatec, São Paulo-SP–Brazil), 2 M NaCl (Dinamica, São Paulo-SP–Brazil), 0.05% spermidine trihydrochloride (Sigma-Aldrich, St. Louis-MO–USA), 2% β-mercaptoethanol (Sigma-Aldrich, St. Louis-MO–USA), (added just before use).

**CTAB protocol 1 (CTAB$_1$).** This CTAB protocol was initially adapted by Zeng and Yang (2002) [29], however, our group slightly modified the protocol to improve sample processing (by using 100 mg of starting material and centrifuged samples at 25,000 ×$g$). Briefly, 900 μL of extraction buffer were heated at 65°C and added to a mortar containing the sample (~100 mg, powdered in liquid nitrogen as described above), then transferred to a 2.0 mL microtube, and vigorously shaken several times. The mixture was incubated at 65°C for 10 min in a shaker. Next, an equal volume of chloroform/isoamyl alcohol (24:1, v/v) (Sigma-Aldrich, St. Louis-MO–USA) was added and the microtube was vigorously mixed. The tube was centrifuged at 10,000 ×$g$ for 10 min at 4°C. The supernatant was recovered and transferred to a new 1.5 mL microtube and re-extracted with an equal volume of chloroform/isoamyl alcohol (~650 μL). The supernatant was slowly and carefully collected to avoid contamination, and another centrifugation was performed at 25,000 ×$g$ for 30 min at 4°C to precipitate and discard the insoluble material. To precipitate the RNA, 0.5 volume of 5 M LiCl (Sigma-Aldrich, St. Louis-MO–USA) was added to the supernatant, mixed well, and the tube was subsequently incubated at 4°C overnight. The RNA pellet was obtained by centrifugation at 25,000 ×$g$ for 30 min at 4°C. The supernatant was discarded, and the pellet was washed three times with 75% (v/v) ethanol (Sigma-Aldrich, St. Louis-MO–USA) and centrifuged at 10,000 ×$g$ for 20 min at 4°C. The pellet was dried at room temperature and solubilized in 30 μL of RNase-free ultrapure water (Macherey-Nagel, Düren–Germany). The extracted RNA was stored in an ULT freezer at -80°C for subsequent analyses.

**CTAB protocol 2 (CTAB$_2$).** This CTAB protocol was adapted by Guzman *et al.* (2014) [42] and further adapted by our group to reduce the initial sample amount to 100 mg of tissue. Moreover, in the precipitation step the LiCl solution concentration was increased (5 M) and incubated for only 4h. Initially, 900 μL of pre-heated extraction buffer (65°C) was added to the mortar containing tissue powder, and the mixture was stirred using a pestle until becoming homogeneous. The mixture was then transferred to a 2 mL microtube and incubated at 65°C for 10 min. Next, an equal volume of chloroform/isoamyl alcohol (24:1, v/v; Sigma-Aldrich, St. Louis-MO–USA) was added and the tube was vigorously mixed. The microtube was centrifuged at 7,000 ×$g$ for 20 min at 4°C. The supernatant was recovered and transferred to a new 1.5 mL microtube, then re-extracted with an equal volume of chloroform/isoamyl alcohol (~650 μL). The supernatant was carefully collected and transferred to a new 1.5 mL microtube. Next, 0.5 volume of 5 M LiCl (Sigma-Aldrich, St. Louis-MO–USA) was added to the supernatant and subsequently incubated at -20°C for 4 hours. The RNA was selectively pelleted by centrifugation at 16,000 ×$g$ for 30 min at 4°C. The pellet was washed with 75% (v/v) ethanol (Sigma-Aldrich, St. Louis-MO–USA) and dried at room temperature. The RNA was solubilized in 30 μL of RNase-free ultrapure water (Macherey-Nagel, Düren–Germany) and stored in an ULT freezer at -80°C for subsequent analyses.

**PureLink RNA Mini Kit (Invitrogen, Carlsbad-CA–USA).** Approximately 100 mg of tissue powdered in liquid nitrogen were used to obtain total RNA. After tissue disruption, 600 μL of the Lysis Buffer (already containing 10 μL of 2-mercaptoethanol for each 1 mL of buffer) was added to promote chemical lysis of the cells. Absolute ethanol (Sigma-Aldrich, St. Louis-

MO–USA; 0.5 volume) was added into the tissue homogenate, and the mixture was vortexed. Next, 700 μL of the lysate was transferred to a silica spin column for the binding of RNA. The column was centrifuged at 12,000 ×g for 15 s. This procedure was carried out until all the lysates had passed through the column and the flow-through was discarded. Subsequently, the column was washed with 700 μL of Wash Buffer I and centrifuged at 12,000 ×g for 15 s to remove the residual buffer. A second wash was performed using 500 μL of Wash Buffer II, and another centrifugation under the same conditions was performed. That last step was again repeated. The column was subjected to additional centrifugation to remove wash buffer residues. The silica spin column was transferred to a new 1.5 mL microtube for RNA elution, in which 30 μL of RNase-free ultrapure water (Macherey-Nagel, Düren–Germany) was used. The obtained RNA was stored in an ULT freezer at -80˚C.

**RNeasy Plant Mini Kit (QIAGEN, Hilden–Germany).** After tissue disruption with liquid nitrogen in a mortar, 450 μL of Buffer RLC (already containing 10 μL of 2-mercaptoethanol for each 1 mL of buffer) were added to the powdered sample (~100 mg), and the mixture was homogenized vigorously. To facilitate tissue disruption, the mixture was incubated at 56˚C for 2 min. The lysate was transferred to a QIA shredder spin column (lilac) and centrifuged at 14,000 ×g for 2 min. The supernatant of the flow-through was transferred to a new 2 mL microtube without disturbing the pellet. Half a volume (~250 μL) of absolute ethanol was added to each sample and the mixture was homogenized by pipetting. Next, ~650 μL of cleared lysate were transferred to the RNeasy Mini spin column (pink) and immediately centrifuged at 8,000 ×g for 15 s. The flow-through was discarded, and the column was washed with 700 μL of Buffer RW1 and centrifuged again at 8,000 ×g for 15 s. The flow-through was discarded and 500 μL of Buffer RPE was added to the column, followed by another centrifugation at 8,000 ×g for 15 s. To remove any residue from the wash buffer, an additional centrifugation at 10,000 ×g for 1 min was performed. The column was then transferred to a new 1.5 mL microtube, and the RNA was eluted from the column by addition of 30 μL of RNase-free ultrapure water (Macherey-Nagel, Düren–Germany) and centrifugation at 8,000 ×g for 1 min. To improve performance, the flow-through was reapplied to the column and again centrifuged. The RNA was stored in an ULT freezer at -80˚C.

**TRIzol method.** For the extraction of total RNA, following the manufacturer's instructions, the leaf tissue (~100 mg) was powdered in a mortar using liquid nitrogen and 1 mL of the TRIzol reagent (Invitrogen, Carlsbad-CA–USA) was added to the sample, which was homogenized using a pestle. The entire mortar content was transferred to a 1.5 mL microtube and incubated for 5 min at room temperature. A centrifugation step at 12,000 ×g for 10 min at room temperature was performed according to the manufacturer's specifications. The whole supernatant was transferred to a new 1.5 mL microtube. Subsequently, 200 μL of chloroform (Sigma-Aldrich, St. Louis-MO–USA) was added to the sample, which was shaken for 30 s and incubated for 5 min at room temperature under a constant slow shaking. The sample was then centrifuged at 12,000 ×g for 15 min at 4˚C. The aqueous phase was transferred to a new 1.5 mL microtube. For RNA precipitation, an equal volume of isopropanol (Sigma-Aldrich, St. Louis-MO–USA; ~500μL) was added to the sample, which was incubated at room temperature for 15 min. After this step, the sample was centrifuged at 12,000 ×g for 10 min at 4˚C and the supernatant was discarded. The pellet was washed using 1 mL of 75% (v/v) ethanol (Sigma-Aldrich, St. Louis-MO–USA). The microtube was centrifuged at 10,000 ×g for 5 min at 4˚C, and the ethanol was completely removed. The pellet was dried at room temperature and the RNA was solubilized in 30 μL of RNase-free ultrapure water (Macherey-Nagel, Düren–Germany). The RNA was stored in an ULT freezer at -80˚C.

**Modified Salzman _et al._ method.** Extraction buffer: 4 M guanidine thiocyanate (LGC, Cotia-SP–Brazil), 100 mM Tris-HCl (Sigma-Aldrich, St. Louis-MO–USA; pH 8.0), 25 mM

sodium citrate pH 8.0 (Dinamica, São Paulo-SP–Brazil), 0.5% N-laurylsarcosine, (Sigma-Aldrich, St. Louis-MO–USA), 1% PVP (mol wt 30,000)(Sigma-Aldrich, St. Louis-MO–USA), 2% β-mercaptoethanol (Sigma-Aldrich, St. Louis-MO–USA; added just before use).

This method was used by Furlan *et al.* (2012) [37] for RNA extraction from *P. guajava* tissue, and was adapted from Salzman *et al.* (1999) [40]. Although this protocol is not fully detailed in the article by Furlan *et al.* (2012) [37], the authors were contacted and cordially provided the detailed protocol used in that study, as described below. Moreover, our group proposed a small adjustment to the initial buffer volume and amount of tissue. Initially, 900 µL of extraction buffer was added to 100 mg of powdered tissue, disrupted using liquid nitrogen in a mortar. After vigorous homogenization, the mixture was transferred to a 2mL microtube and incubated for 1 min at room temperature. Subsequently, 750 µL of chloroform/isoamyl alcohol (24:1, v/v; Sigma-Aldrich, St. Louis-MO–USA) was added and the sample was mixed slowly by inversion for 10 min. Next, the sample was centrifuged at 16,000 ×*g* for 10 min at 4˚C. The aqueous phase (~600 µL) was transferred to a new 1.5 mL microtube and re-extracted with an equal volume of chloroform/isoamyl alcohol. After centrifugation under the same conditions as described above, the aqueous phase was recovered and transferred to a new 1.5-mL microtube. The RNA was then precipitated by the addition of 2 volumes of cold absolute ethanol (Sigma-Aldrich, St. Louis-MO–USA) and 0.1 volume of 5 M NaCl (Dinamica, São Paulo-SP–Brazil) at least 3 h. The sample was centrifuged at 16,000 ×*g* for 10 min at 4˚C and the formed pellet was washed with 75% ethanol (Sigma-Aldrich, St. Louis-MO–USA). The RNA was solubilized in 800 µL of RNase-free ultrapure water and an equal volume of phenol/chloroform/isoamyl alcohol (25:24:1, v/v) (Sigma-Aldrich, St. Louis-MO–USA) was added. The mixture was homogenized for 5 min at room temperature, then centrifuged at 12,000 ×*g* for 10 min at room temperature. The aqueous phase was recovered and transferred to a new 1.5-mL microtube. The RNA was precipitated by addition of 2 volumes of ice-cold absolute ethanol (Sigma-Aldrich, St. Louis-MO–USA) and 0.1 volume of 5M NaCl (Dinamica, São Paulo-SP–Brazil), with an overnight incubation at -20˚C. The samples were centrifuged at 16,000 ×*g* for 10 min at 4˚C, and the formed pellet was washed with 75% (v/v) ethanol (Sigma-Aldrich, St. Louis-MO–USA). The pellet was dried at room temperature, and the RNA was solubilized in 30 µL of RNase-free ultrapure water (Macherey-Nagel, Düren–Germany) and stored in an ULT freezer at -80˚C for subsequent analyses.

## RNA analysis (yield, purity, and integrity)

RNA purity and concentration were assessed by determining the absorbance of RNA in RNase-free water at 230, 260, and 280 nm, using a NanoDrop ND-1000 spectrophotometer (Thermo Scientific, USA). The RNA yield was calculated based on absorbance at 260 nm ($A_{260}$). The ratios $A_{260}/A_{280}$ and $A_{260}/A_{230}$ were assessed to evaluate RNA purity. RNA integrity was evaluated from the 28S and 18S rRNA bands in 1.0% (w/v) formaldehyde–agarose gel after electrophoresis, staining with 1:20,000 GelRed (Biotium, Fremont-CA–USA) and visualization with Gel Doc XR+ System (Bio-Rad, USA). Quality analysis of ten RNA samples were assessed by TapeStation (Agilent Technologies, Santa Clara-CA–USA) using RNA ScreenTape assay, after a DNase I treatment and the RIN (RNA Integrity Number) were calculated.

## cDNA synthesis and reverse transcription polymerase chain reaction (RT-qPCR)

Initially, the RNA was treated with RQ1 RNase-Free DNase I (Promega, USA) following the manufacturer's instructions. The first-strand cDNA was synthesized from 3.0 µg of the total RNA by reverse transcriptase with random primer (50ng/µL), according to instructions of the

SuperScript™ IV First-Strand Synthesis System (Invitrogen, Carlsbad-CA–USA). The PCR reaction mix (20 μL) was prepared following instructions of the PowerUp SYBR$^{®}$ Green Master Mix (Life Technologies, USA) in a LightCycler$^{®}$ 96 System (Roche Life Science, Germany) thermal cycler. The cycling conditions were as follows: initial denaturation at 95˚C for 2 min; 40 cycles at 95˚C for 15 s and 72˚C for 60 s; and standard dissociation step from 50 to 95˚C with increments of 0.5˚C for 10 s. Real-time PCR analyses were performed to evaluate the quality of the RNA samples from *P. guajava* tissues obtained using the CTAB$_2$ protocol. Amplification of the samples was performed using a common endogenous gene, Histone H2A, forward primer 5′-AAGCCGGTCTCTCGGTCTGT-3′, reverse primer 5′-GCATTA CCAGCCAACTCCAG-3′ (S3 Fig). The primer pair was designed from a gene sequence obtained after analyzing the orthologous genes of the H2A gene of *Eugenia uniflora* (species of the Myrtaceae family) using a draft of the *P. guajava* genome obtained by our laboratory (unpublished data).

## Inhibition test

Quantitative analysis was performed to evaluate the presence of inhibitors in samples from *P. guajava* leaves obtained using the tested CTAB protocols (CTAB$_1$ and CTAB$_2$). For this test, RNA was treated with RQ1 RNAse-Free DNAse I (Promega, USA) following the manufacturer's instructions. The first cDNA strand was synthesized from 5.0 μL of the total RNA by reverse transcriptase with random primer (50ng/μL), according to the instructions of the SuperScript™ III First-Strand Synthesis System (Invitrogen, Carlsbad-CA–USA). RT-qPCR amplification was performed using a common endogenous gene, Histone H2A. Standard curves were constructed foreach sample, in which the largest volume of cDNA was 1μL (~100ng per reaction) and the lowest was 0.125 μL (~12.5ng per reaction), following a dilution factor of 1:2. From the Ct values obtained for each dilution, a graph was plotted relating Ct vs. log (cDNA volume), and the linear equation was obtained for each sample. Using the Ct value of the lowest dilution, a curve was estimated in which the efficiency would be optimal, increasing one Ct value per cycle.

## Supporting information

**S1 Fig. Analysis of RNA samples from *P. guineense*.** Table summarizing the quantification of the RNA samples in a NanoDrop spectrophotometer (Thermo Fischer, USA), as well as the ratios A$_{260}$/A$_{280}$ and A$_{260}$/A$_{230}$. Denaturing agarose gel (1%) electrophoresis of total extracted RNA (2 μL) stained with GelRed (Biotium, USA), original image.
(DOCX)

**S2 Fig. Images exemplification of tissues from *P. guajava* used in this study.** Five different tissues from *P. guajava* used to obtain the total RNA. Flower bud (FB); Immature leaf (IL); Young leaf (YL); Mature leaf (ML); and Root (R).
(DOCX)

**S3 Fig. H2A primer efficiency.** For the primer efficiency analysis, standard curves were constructed with cDNA quantities of 50 ng (highest), 25 ng, 2.5 ng, 0.25 ng and 0.025 ng (lowest). The efficiency was 98%.
(DOCX)

**S4 Fig. Original RNA agarose gel electrophoresis corresponding to Fig 1.** 1-Ladder (faint); 2-Positive control (PureLink RNA Kit); 3- Sample1 (PureLink RNA Kit); 4- Sample2 (Pure-Link RNA Kit); 5- empty; 6- Positive control (RNeasy Plant Kit); 7- Sample1 (RNeasy Plant Kit); 8- Sample2 (RNeasy Plant Kit); 9- empty; 10- Positive control (CTAB$_1$); 11- Sample1

(CTAB$_1$); 12- Sample2 (CTAB$_1$); 13- empty; 14- Positive control (CTAB$_2$); 15- Sample1 (CTAB$_2$); 16- Sample2 (CTAB$_2$); 17- empty; 18- Positive control (TRIzol); 19- Sample1 (TRIzol); 20- Sample2 (TRIzol); 21- empty; 22- Positive control (Guanidine prorocol); 23- Sample1 (Guanidine prorocol); 24- Sample2 (Guanidine prorocol). Note that in Fig 1, we inverted the presentation of the results of the CTAB protocols with that of the commercial kits, to be more consistent with the presentation in the text. Therefore, we are presenting a cropped figure.
(DOCX)

**S5 Fig. Original RNA agarose gel electrophoresis corresponding to Fig 2A.** Cortibel Samples 1- Ladder (omitted); 2- Immature leaf1; 3- Immature leaf2; 4- Young leaf1; 5- Young leaf2; 6- Mature leaf1; 7- Mature leaf2; 8- Root1; 9- Root2; 10- Flower bud1; 11- Flower bud2; 12- Flower bud3(omitted). Note that in Fig 2A, we inverted the presentation of the results of the flower bud samples.
(DOCX)

**S6 Fig. Original RNA agarose gel electrophoresis corresponding to Fig 2A.** Paluma Samples 1- Ladder (omitted); 2- Immature leaf1; 3- Immature leaf2; 4- Young leaf1; 5- Young leaf2; 6- Mature leaf1; 7- Mature leaf2; 8- Root1; 9- Root2; 10- empty.
(DOCX)

**S1 Table. RNA yield and purity of samples used in qRT-PCR inhibition test.**
Table summarizing the quantification of the RNA samples in a NanoDrop spectrophotometer, as well as the ratios $A_{260}/A_{280}$ and $A_{260}/A_{230}$.
(DOCX)

**S1 File.**
(PDF)

## Author Contributions

**Conceptualization:** Paola de Avelar Carpinetti, Adésio Ferreira, Marcia Flores da Silva Ferreira.

**Data curation:** Vinicius Sartori Fioresi, Marcia Flores da Silva Ferreira.

**Formal analysis:** Paola de Avelar Carpinetti, Vinicius Sartori Fioresi.

**Funding acquisition:** Adésio Ferreira.

**Investigation:** Paola de Avelar Carpinetti, Vinicius Sartori Fioresi, Thais Ignez da Cruz, Francine Alves Nogueira de Almeida, Drielli Canal.

**Methodology:** Paola de Avelar Carpinetti, Vinicius Sartori Fioresi, Thais Ignez da Cruz, Francine Alves Nogueira de Almeida, Drielli Canal, Marcia Flores da Silva Ferreira.

**Project administration:** Adésio Ferreira.

**Resources:** Adésio Ferreira.

**Supervision:** Marcia Flores da Silva Ferreira.

**Visualization:** Paola de Avelar Carpinetti.

**Writing – original draft:** Paola de Avelar Carpinetti, Vinicius Sartori Fioresi.

**Writing – review & editing:** Paola de Avelar Carpinetti, Marcia Flores da Silva Ferreira.

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
