## [Decision Letter · Decision Letter 0]

8 Apr 2021

PONE-D-21-07735

Efficient method for isolation of high-quality RNA from Psidium guajava L. tissues

PLOS ONE

Dear Dr. Ferreira,

Thank you for submitting your manuscript to PLOS ONE. After careful consideration, we feel that it has merit but does not fully meet PLOS ONE’s publication criteria as it currently stands. Therefore, we invite you to submit a revised version of the manuscript that addresses the points raised during the review process.

In addition to the comments raised by the reviewers', please do the following:

1- Full vendor details should include company, city (state), and country. Please check and amend throughout the text

2- Please don't use "We, our". Use impersonal phrasing throughout the text

3- Fig. 3 is not good, redo it in color

4- Before conclusion please add these two section

a- Comparisons with other methods: You should compare your finding with other techniques to prove the applicability, if any. Summarize the findings in Table

b- "Study strength and limitations". In this section, you have to discuss the strength and limitations of this study

5- Proofread the text for grammar and syntax errors

We look forward to receiving your revised manuscript.

Kind regards,

A. M. Abd El-Aty

Academic Editor

PLOS ONE

Journal Requirements:

PLOS ONE now requires that authors provide the original uncropped and unadjusted images underlying all blot or gel results reported in a submission’s figures or Supporting Information files. This policy and the journal’s other requirements for blot/gel reporting and figure preparation are described in detail at https://journals.plos.org/plosone/s/figures#loc-blot-and-gel-reporting-requirements and https://journals.plos.org/plosone/s/figures#loc-preparing-figures-from-image-files. When you submit your revised manuscript, please ensure that your figures adhere fully to these guidelines and provide the original underlying images for all blot or gel data reported in your submission. See the following link for instructions on providing the original image data: https://journals.plos.org/plosone/s/figures#loc-original-images-for-blots-and-gels.

PLOS requires an ORCID iD for the corresponding author in Editorial Manager on papers submitted after December 6th, 2016. Please ensure that you have an ORCID iD and that it is validated in Editorial Manager. To do this, go to ‘Update my Information’ (in the upper left-hand corner of the main menu), and click on the Fetch/Validate link next to the ORCID field. This will take you to the ORCID site and allow you to create a new iD or authenticate a pre-existing iD in Editorial Manager. Please see the following video for instructions on linking an ORCID iD to your Editorial Manager account: https://www.youtube.com/watch?v=_xcclfuvtxQ

Reviewers' comments:

Reviewer's Responses to Questions

**Comments to the Author**

1. Is the manuscript technically sound, and do the data support the conclusions?

Reviewer #1: Yes

Reviewer #2: Yes

Reviewer #3: Yes

2. Has the statistical analysis been performed appropriately and rigorously? 

Reviewer #1: No

Reviewer #2: Yes

Reviewer #3: Yes

3. Have the authors made all data underlying the findings in their manuscript fully available?

Reviewer #1: Yes

Reviewer #2: Yes

Reviewer #3: Yes

4. Is the manuscript presented in an intelligible fashion and written in standard English?

Reviewer #1: Yes

Reviewer #2: Yes

Reviewer #3: Yes

5. Review Comments to the Author

Reviewer #1: The authors have addressed most of the issues raised during the first review process. In order to clearly evaluate the efficiency of the different extraction protocols, however, statistical analysis should be conducted on the data in Table 1 and Table 2 (i.e the authors should evaluate and indicate if there is any significant difference in yield in relation to variation in extraction protocol, tissue type, and genotype). The result and discussion parts should also describe these statistical analysis results.

Reviewer #2: The manuscript is technically sound and all the suggestions given by the reviewers has been addressed and necessary corrections has been done to improve the manuscript. The data presented in the manuscript is sufficient and well analyzed as per the objective of the experiments. All the data are available in the manuscript and its supplementary part. There is no additional comments to authors.

Reviewer #3: The authors responded adequately to the questions posed and have improved the manuscript, for which I consider it convenient for its publication in this journal.

Minor comments

Line 116, 129, 142, 146. Since the authors added this reference in the Results and Discussion section, they must add the reference number.

Line 235-237. True, the authors demonstrated to obtain high-quality RNA, however, in this section it could be convenient (if the authors consider) to add that this RNA obtained may still have limitations for other molecular studies such as gene expression analysis by RNA-seq (among other) , due to the RIN values obtained, and for the reasons mentioned in previous paragraphs. Examples of Ref. (Doi: 10.1186 / 1471-2199-7-3 and https://doi.org/10.1016/j.mimet.2020.105855)

6. PLOS authors have the option to publish the peer review history of their article (what does this mean?). If published, this will include your full peer review and any attached files.

Reviewer #1: No

Reviewer #2: **Yes: **Dr. Rakesh Singh

Reviewer #3: No

---

## [Author Response · Author response to Decision Letter 0]

25 May 2021

Academic Editor’s comment:

1- Full vendor details should include company, city (state), and country. Please check and amend throughout the text.

Answer: We appreciate the recommendation and we have corrected it in the manuscript, as highlighted in the text.

2- Please don't use "We, our". Use impersonal phrasing throughout the text.

Answer: We appreciate the recommendation and we have added the information in the manuscript, as highlighted in the text.

3- Fig. 3 is not good, redo it in color.

Answer: We appreciate the recommendation and we have corrected it.

4- Before conclusion please add these two section:

a- Comparisons with other methods: You should compare your finding with other techniques to prove the applicability, if any. Summarize the findings in Table.

Answer: We appreciate the editor's suggestion; however, we believe that this information would be most appropriate if placed immediately after the results of comparison of protocols. Thus, we have included in this revised version the new suggested table (table 2 - line 163).

b- "Study strength and limitations". In this section, you have to discuss the strength and limitations of this study.

Answer: We appreciate the editor's suggestion. We have included this section in the revised version (lines 321-340).

5- Proofread the text for grammar and syntax errors.

Answer: We sent the manuscript back for editing by a native speaker to correct grammar and syntax errors. We believe that after this review the manuscript will be presented in an intelligible form and written in standardized English.

Journal Requirements:

 Answer: We have corrected it in the revised manuscript.

PLOS ONE now requires that authors provide the original uncropped and unadjusted images underlying all blot or gel results reported in a submission’s figures or Supporting Information files. When you submit your revised manuscript, please ensure that your figures adhere fully to these guidelines and provide the original underlying images for all blot or gel data reported in your submission. 

Answer: The gel raw images data are in Supporting Information.docx file and Raw images.pdf file.

3. PLOS requires an ORCID iD for the corresponding author in Editorial Manager on papers submitted after December 6th, 2016. Please ensure that you have an ORCID iD and that it is validated in Editorial Manager.

 Answer: The ORCID iD of the corresponding author is validated.

Reviewer #1’s comment:

The authors have addressed most of the issues raised during the first review process. In order to clearly evaluate the efficiency of the different extraction protocols, however, statistical analysis should be conducted on the data in Table 1 and Table 2 (i.e the authors should evaluate and indicate if there is any significant difference in yield in relation to variation in extraction protocol, tissue type, and genotype). The result and discussion parts should also describe these statistical analysis results.

Answer: We appreciate and agree with the reviewer's comments, and modified tables 1 and 2, including statistical analysis, as well as adding this information in the discussion (lines 155-162 and 223-231).

Reviewer #2’s comment:

The manuscript is technically sound and all the suggestions given by the reviewers has been addressed and necessary corrections has been done to improve the manuscript. The data presented in the manuscript is sufficient and well analyzed as per the objective of the experiments. All the data are available in the manuscript and its supplementary part. There is no additional comments to authors.

Answer: We appreciate the comments.

Reviewer #3’s comment:

The authors responded adequately to the questions posed and have improved the manuscript, for which I consider it convenient for its publication in this journal.

Minor comments

Line 116, 129, 142, 146. Since the authors added this reference in the Results and Discussion section, they must add the reference number.

Answer: We appreciate the recommendation and we have added the information in the manuscript.

Line 235-237. True, the authors demonstrated to obtain high-quality RNA, however, in this section it could be convenient (if the authors consider) to add that this RNA obtained may still have limitations for other molecular studies such as gene expression analysis by RNA-seq (among other) , due to the RIN values obtained, and for the reasons mentioned in previous paragraphs. Examples of Ref. (Doi: 10.1186 / 1471-2199-7-3 and https://doi.org/10.1016/j.mimet.2020.105855).

Answer: We appreciate the recommendation and we have added the information in the manuscript (lines 244-255).

---

## [Decision Letter · Decision Letter 1]

2 Jun 2021

PONE-D-21-07735R1

Efficient method for isolation of high-quality RNA from Psidium guajava L. tissues

PLOS ONE

Dear Dr. Ferreira,

Thank you for submitting your manuscript to PLOS ONE. After careful consideration, we feel that it has merit but does not fully meet PLOS ONE’s publication criteria as it currently stands. Therefore, we invite you to submit a revised version of the manuscript that addresses the points raised during the review process.

ACADEMIC EDITOR: Still the MS needs language editing, in particular, the parts which are added in the last round of revision. 

We look forward to receiving your revised manuscript.

Kind regards,

A. M. Abd El-Aty

Academic Editor

PLOS ONE

Journal Requirements:

Reviewers' comments:

Reviewer's Responses to Questions

**Comments to the Author**

1. If the authors have adequately addressed your comments raised in a previous round of review and you feel that this manuscript is now acceptable for publication, you may indicate that here to bypass the “Comments to the Author” section, enter your conflict of interest statement in the “Confidential to Editor” section, and submit your "Accept" recommendation.

Reviewer #1: All comments have been addressed

2. Is the manuscript technically sound, and do the data support the conclusions?

Reviewer #1: Yes

3. Has the statistical analysis been performed appropriately and rigorously? 

Reviewer #1: Yes

4. Have the authors made all data underlying the findings in their manuscript fully available?

Reviewer #1: Yes

5. Is the manuscript presented in an intelligible fashion and written in standard English?

Reviewer #1: Yes

6. Review Comments to the Author

Reviewer #1: The authors have fully addressed the issues raised during the two rounds of review processes. Hence, I recommend the acceptance of the manuscript in its current status.

7. PLOS authors have the option to publish the peer review history of their article (what does this mean?). If published, this will include your full peer review and any attached files.

Reviewer #1: No

---

## [Author Response · Author response to Decision Letter 1]

9 Jul 2021

In our last contact, all the points raised by the reviewers had been satisfactorily answered and all of them approved the manuscript. Lastly, the Academic Editor informed that language editing was still necessary.

 In this final version, we have joined efforts to ensure that the adjustments were finally met and requested an additional Academic editing service, as can be seen by the two certificates attached to the submission files (Certificates of English Editing.pdf).

 We hope the Academic Editor agrees that we have addressed the issues raised by the reviewers, and the manuscript is now suitable for publication in Plos One.

Sincerely, 

Marcia Flores da Silva Ferreira

Professor at Universidade Federal do Espírito Santo

ZIP: 29.500-000 Alegre – ES, Brazil

E-mail: marcia.ferreira@ufes.br

---

## [Editor Report · Decision Letter 2]

13 Jul 2021

Efficient method for isolation of high-quality RNA from Psidium guajava L. tissues

PONE-D-21-07735R2

Dear Dr. Ferreira,

We’re pleased to inform you that your manuscript has been judged scientifically suitable for publication and will be formally accepted for publication once it meets all outstanding technical requirements.

Kind regards,

A. M. Abd El-Aty

Academic Editor

PLOS ONE
---

## [Editor Report · Acceptance letter]

16 Jul 2021

PONE-D-21-07735R2 

Efficient method for isolation of high-quality RNA from *Psidium guajava* L. tissues 

Dear Dr. Ferreira:

I'm pleased to inform you that your manuscript has been deemed suitable for publication in PLOS ONE. Congratulations! Your manuscript is now with our production department. 

Kind regards, 

on behalf of

Prof. A. M. Abd El-Aty 

Academic Editor

PLOS ONE